# Development of the fetal myocardium and changes in myocardial fibers orientation

**Biagio Castaldi**[1]*, **Marny Fedrigo**[2], **Daniela P. Boso**[3], **Irene Cattapan**[1],
**Elena De Filippi**[3], **Francesca M. Susin**[3], **Paolo Peruzzo**[3], **Giulia Comunale**[3],
**Paola Veronese**[1], **Ornella Milanesi**[1], **Annalisa Angelini**[2], **Giovanni Di Salvo**[1]

1 Department of Women's and Children's Health, University of Padua, Padova, Italy, 2 Department of Cardiac Thoracic and Vascular Sciences and Public Health, University of Padua, Padova, Italy, 3 Department of Civil Environmental and Architectural Engineering, University of Padua, Padova, Italy

* b.castaldi@yahoo.it

## Abstract

### Background and aims

The mature left ventricular myocardium is arranged in a complex three-dimensional network of fibers that form a counterclockwise helix in the endocardial layer and a clockwise helix in the epicardial layer. There are no data in the literature on the development of left ventricular myocardium during the fetal life. The aims of this paper were to study the physiological maturation steps of the LV myocardium in fetuses from 17 to 40 gestational weeks, by means of speckle tracking applied to the endocardial and epicardial aspect of the left ventricle, and, to confirm our finds, through the histologic study of the myocardium of demised fetuses.

### Methods and results

We studied longitudinal endocardial and epicardial strain by echocardiography in 105 fetuses. Twenty non-diseased fetal hearts from autopsies were selected to assess the layer thickness and cardiac fiber orientation in relation to gestational age. Echocardiography showed a progressive increasing of epicardial/endocardial longitudinal strain ratio with gestational age (r=0.51; p<0.0001). The strain rate E/A ratio increased over time (r=0.27; p=0.018). Histological data revealed that during the same gestational period, the proportion of the epicardial layer increased fourfold, the mesocardiac layer decreased and the endocardial layer remained stable. We found an excellent correlation between the epicardial to endocardial strain ratio and epicardial to endocardial wall thickness (r=0.950, p<0.001).

### Conclusions

Left ventricular myocardium maturation begins early during fetal life. As the fetus develops, both the relative tissue volume and peak systolic strain rates shift together from the endocardium towards the epicardium. It is a slow process, completed late in fetal life.

**Data availability statement:** All relevant data were added on a supporting information file

**Funding:** The authors received no specific funding for this work.

**Competing interests:** The authors have declared that no competing interests exist.

**Abbreviations:** BMI, body mass index; ECG, electrocardiogram; GW, gestational weeks; LV, left ventricle; MRI, magnetic resonance imaging; PAIVS, pulmonary atresia with intact ventricular septum; STE, Speckle tracking echocardiography; TDI, tissue Doppler imaging.

## Introduction

The left ventricular myocardium is arranged in a complex three-dimensional network of fibers that form a counterclockwise helix in the endocardial layer and a clockwise helix in the epicardial layer [1]. This peculiar organization is fundamental to myocardial functionality because the left ventricular contraction is both longitudinally and radially oriented, and characterized by ventricular twisting [2]. This architecture may change in congenital and acquired heart diseases [3,4].

Whereas the embryological development of the human heart is well documented, much less is known about fetal myocardial development and the mechanism underlying this process. Using fetal magnetic imaging, Mekkaoui et al. [5] demonstrated that the myocardial fiber orientation of the left ventricle changes during gestation, however these data were not validated by histological investigations or confirmed by a similar study [5]. On the other hand, it has been demonstrated that myocardial function changes in the first days after birth thereby promoting changes of the left ventricular twist and longitudinal strain [6,7].

With the advent of speckle-tracking echocardiography (STE), endocardial and epicardial function can now be assessed separately *in vivo* [8,9]. Although strain Doppler and STE are widely used to study the left ventricular function in fetal hearts [10–12], no study has yet evaluated the left ventricular longitudinal strain in the two layers separately.

The aims of this study were to assess *in vivo* the endocardial and epicardial longitudinal strain in fetuses of different gestational ages by STE as surrogate of layer maturation, and to confirm, on histological sections, the relative wall thickness and fiber orientation of the endocardial and epicardial layers at different gestational ages.

## Methods

Informed consent was obtained from each patient or their parents. The study protocol conforms to the ethical guidelines (Declaration of Helsinki 1975) and approved by the Institution's human research committee. The data that support the findings of this study are available from the corresponding author upon reasonable request.

This study consists of two parts: *in vivo* study, and histological evaluation of normal fetal hearts. One hundred and five fetuses between 17 and 40 gestational weeks were enrolled in the *in vivo* study. They were selected among healthy singleton pregnancies undergoing general obstetrical ultrasound screening, and they agreed to undergo fetal echocardiography voluntarily. All fetuses examined were free from heart disease and extracardiac pathology, demonstrated normal intrauterine growth and had a normal newborn examination. The pregnant women were otherwise healthy and on no medications other than vitamins. In particular, women with known hypertension, auto-immune diseases, obesity (BMI>30 kg/m$^2$ before pregnancy), hypertension during pregnancy or previous pregnancies complicated by gestational diabetes, hypertension, pre-eclampsia or eclampsia were excluded from the study. Maternal diabetes was excluded by oral glucose tolerance test performed between the 26th and 28th gestational weeks in all the patients. The normal heart anatomy and function were confirmed after the birth. Neonatological evaluation was unremarkable in all the fetuses selected. The acoustic window and fetal position were ideal for STE assessment: the image quality was good, and the left ventricle was acquired in vertical position. The long axis of the ventricle formed an angle with the probe vertical line of -30/+30°. The sector width was adjusted to obtain the highest frame rate possible (between 90 and 150 fps)

For the histological sections, we obtained 20 hearts without any cardiac diseases from fetal autopsies (gestation age: 13–40 week) available in the Veneto Region Registry of Cardiovascular Pathology Unit of our Hospital. Fetuses were referred to our Cardiovascular Pathology Department for post-mortem investigation in 2015 and 2016. Indications for autopsy were:

miscarriage (4 cases), voluntary abortion for social reasons (4 cases), intrauterine death (11 cases) and perinatal death (1 case). We excluded cases with intrauterine growth restriction and cases with placental malperfusion pathologies. According to fetal anthropometric indexes these hearts were within a normal range for the gestational weeks.

## Echocardiography

Speckle-tracking was performed on apical 4-chamber views. We used a GE Vivid E9 imaging ultrasound system (General Electric, Waukesha, WI, USA), the 5-MSc and 6S probes. Images were processed offline with a commercially available software (EchoPac v. 11.2, General Electric, Waukesha, WI, USA). To evaluate fetal hearts, we gated the cardiac cycle on the mitral valve opening and closure, while we used ECG in neonates. The endocardial and epicardial borders of the left ventricular wall were traced in an end-systolic frame. The software automatically selects six equidistant tissue-tracking regions of interest in the myocardium (manually adjustable if needed). To measure strain, the myocardium was automatically divided into endocardial and epicardial layers (not further modifiable). To measure strain rate, we recorded the mean S, E and A peak values and the E/A ratio. Visual control of tracking quality was performed and optimized if required, by adjusting the region of interest or manually correcting the contour to ensure adequate automatic tracking. All data were acquired at a frame rate from 90 to 150 frames per second (mean 110 fps, between 45 and 55 frames for cardiac cycle). Strain and strain rate values were obtained from three consecutive cardiac cycles.

Left ventricular volume and mass were calculated applying the Cylinder hemi-ellipsoid Bullet method using the apical 4 chambers view [13]. We also recorded the mean left ventricle end diastolic wall thickness, the left ventricular end diastolic diameter, the left ventricular longitudinal end diastolic diameter and the left ventricular longitudinal to transverse length ratio. Interobserver variability, expressed as coefficient of variation, was assessed by two independent investigators who analyzed 20 randomly chosen cine-loops. For intra-observer variability, 10 cine loops were analyzed by the same investigator twice within an interval of 4 weeks. The second round of intra-observer measures was blinded to results from initial measures.

## Histology

A normal heart anatomy was confirmed in all the specimens selected. For each heart we cut two sequential blocks at middle-basal left ventricle free wall one longitudinal and one transverse. The samples were included in paraffin, cut into 5 μm slices, and then stained with Azan-Mallory for measurement of layer thickness and myofibers orientation. We acquired images of each histological section using a Zeiss Video camera connected to a light microscopy (Zeiss Camera, Axio Com color 412–312) and a computer supporting Image Pro Plus morphometric software v. 6.0 (Media Cybernetics, Silver Spring, MA). Longitudinal sections were evaluated to quantify each layer's thickness, while transversal sections were analyzed to determine the orientation of myofibers.

Longitudinal sections were captured at a magnification of 2.5x and the program was calibrated. Each layer thickness was assessed according to fiber orientation: starting from the non-compacted concave part we defined the innermost longitudinal fibers as endocardium, the medial perpendicularly-cut fibers as mesocardium and the outermost longitudinal fibers as epicardium (Fig 1). The layers were identified according to different orientation of the fibers: vertically cut fibers of middle myocardium appeared lighter in color, because of the major amount of Orange G staining cytoplasm, making it possible to recognize them from the other layers. In the section it was therefore possible to clearly distinguish the longitudinal fibers of the endocardium and epicardium, colored purple red, and the fibers of the mesocardium, which

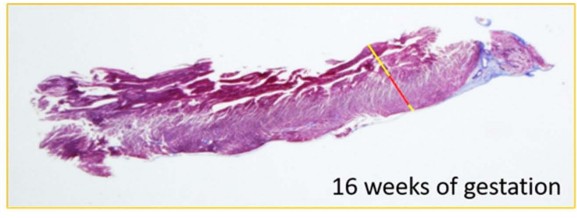

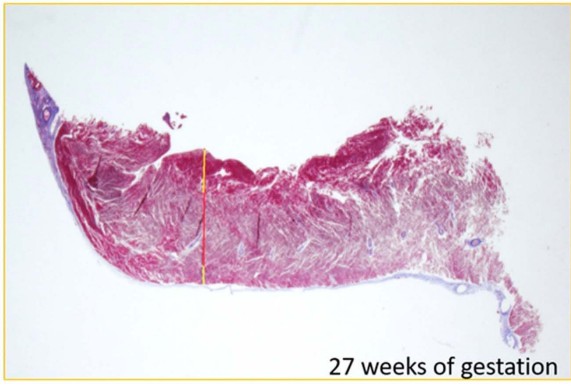

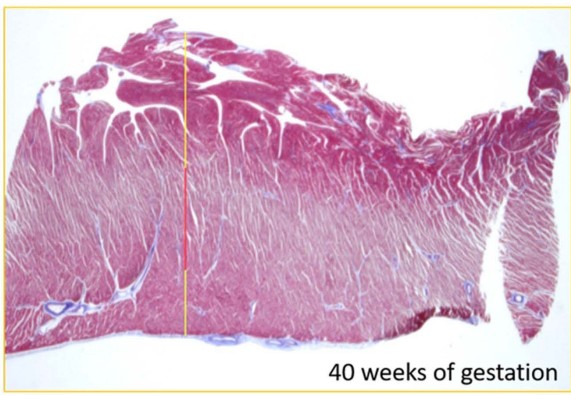

**Fig 1. Sagittal slices for assessment of layer thickness and fiber's orientation. On the top, specimen in a 16 weeks of gestation fetus: the longitudinal endocardial layer is well represented (yellow line), the radial fibers are about 40% of the total wall thickness (red line), while the longitudinal epicarial layer was thin. The epicardial longitudinal layer progressively increased over the gestational weeks (in the middle, 27 weeks of gestation fetus, bottom, 40 weeks of gestation fetus).**

presented a lighter shade as well as a different pattern. Using the same software, we were able to trace line segments for which the program gave us numerical values in micrometer (μm). We measured total wall thickness, non-compacted thickness, endocardial thickness, middle myocardium thickness and epicardial thickness. We carefully avoided artifacts. The measurements were repeated 5 times for each level and in different parts of the specimen. We recorded the mean value for each layer. In addition, the relative layer thickness was calculated.

To study the fibers' angulation, we applied Streeter's methodology to transversal sections (Fig 2) [14,15]. Transversal sections were acquired through consecutive sampling moving from the inner side trough the outer one. Magnification used was 20x and morphometric software was properly calibrated. Every sampling area was divided into seven equal regions of interest [16]. Using the function "measurements" we reported the mean orientation of myocardial fibers considering the angle formed between fibers and plan of section for each layer. Then, we averaged the data collected for each layer.

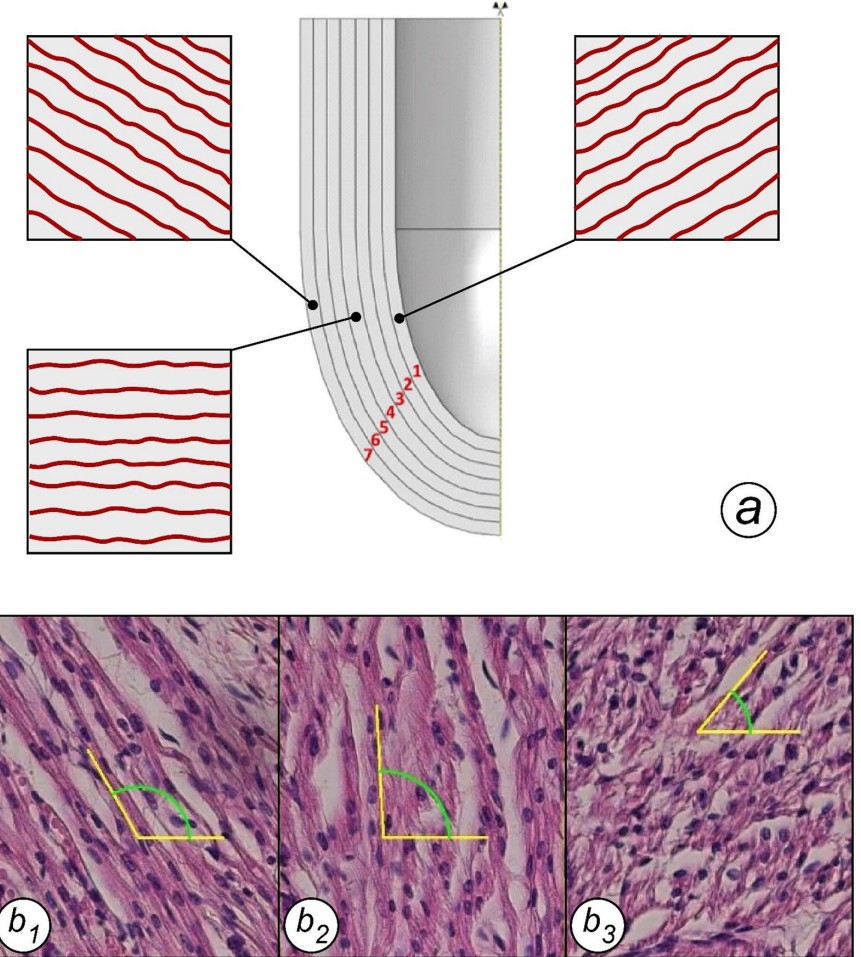

**Fig 2. The ventricular myocardium was divided into seven layers to model the transmural variation of fiber orientation (A). Each layer contains fibers oriented according to the angles measured by histology across the myocardium thickness (b1: epicardial layer, b2: mesocardium, b3: endocardial layer). Since fiber angles vary with gestational age, three models were developed for 20, 30 and 40-week-old fetuses (see Figs 7 and 8).**

After histopathological assessment, the slices were examined by a pathologist who was blinded to previous findings.

## Statistical analysis

The distribution of echocardiographic data was normal as assessed by the Kolmogorov–Smirnov test. Quantitative values are reported as mean ±1 Standard Deviation. The correlation between echo data and gestational age was evaluated with the Pearson test and regression plots, including the research of the more appropriate regression model (linear, quadratic, cubic, logarithmic, logistic, and exponential). The correlation between epicardial to endocardial ratio assessed by speckle tracking and histology was performed by using Spearman's Rank Test. Data were clusterized for gestational week. Thus, we calculated the mean echocardiographic epi/endo ratio and compared echo data with histological data at any gestational week. The null hypothesis was rejected for a P-value <0.05. All analyses were performed using a commercially available package (SPSS, Rel 18.0 2009. Chicago: SPSS Inc.).

# Results

## Echocardiography

Table 1 shows the mean values of longitudinal epicardial and endocardial strain at different gestational ages. Endocardial strain, from the 24th week of gestation, progressively decreased with gestational age (r=0.35; p=0.001), whereas epicardial strain remained substantially unchanged (r=0.04; p=0.698). Therefore, the epicardial to endocardial longitudinal strain ratio progressively increased (r=0.51; p<0.0001; y=0.007x+0.48) (Fig 3. As shown in Fig 4, the S

**Table 1. Endocardial and epicardial longitudinal strain and epicardial to endocardial longitudinal strain ratio at different gestational ages [in the square bracket the sample size]. Endocardial strain: r=0.35; p=0.001. Epicardial strain: r=0.04; p=0.698. Epicardial to Endocardial strain ratio (Epi/Endo): r=0.51; p<0.0001.**

| Gestational weeks | Endocardial S | Epicardial S | Epi/Endo Ratio |
|---|---|---|---|
| 17.0-23.0 [n=34] | -27.48±5.98 | -17.11±4.01 | 0.623±0.074 |
| 23.1-28.0 [n=22] | -27.05±5.17 | -17.67 ±4.55 | 0.652±0.104 |
| 28.1-33.0 [n=29] | -26.53±3.90 | -18.15±3.33 | 0.681±0.067 |
| 33.1-40.0 [n=20] | -24.82±3.68 | -18.60±2.54 | 0.748±0.036 |

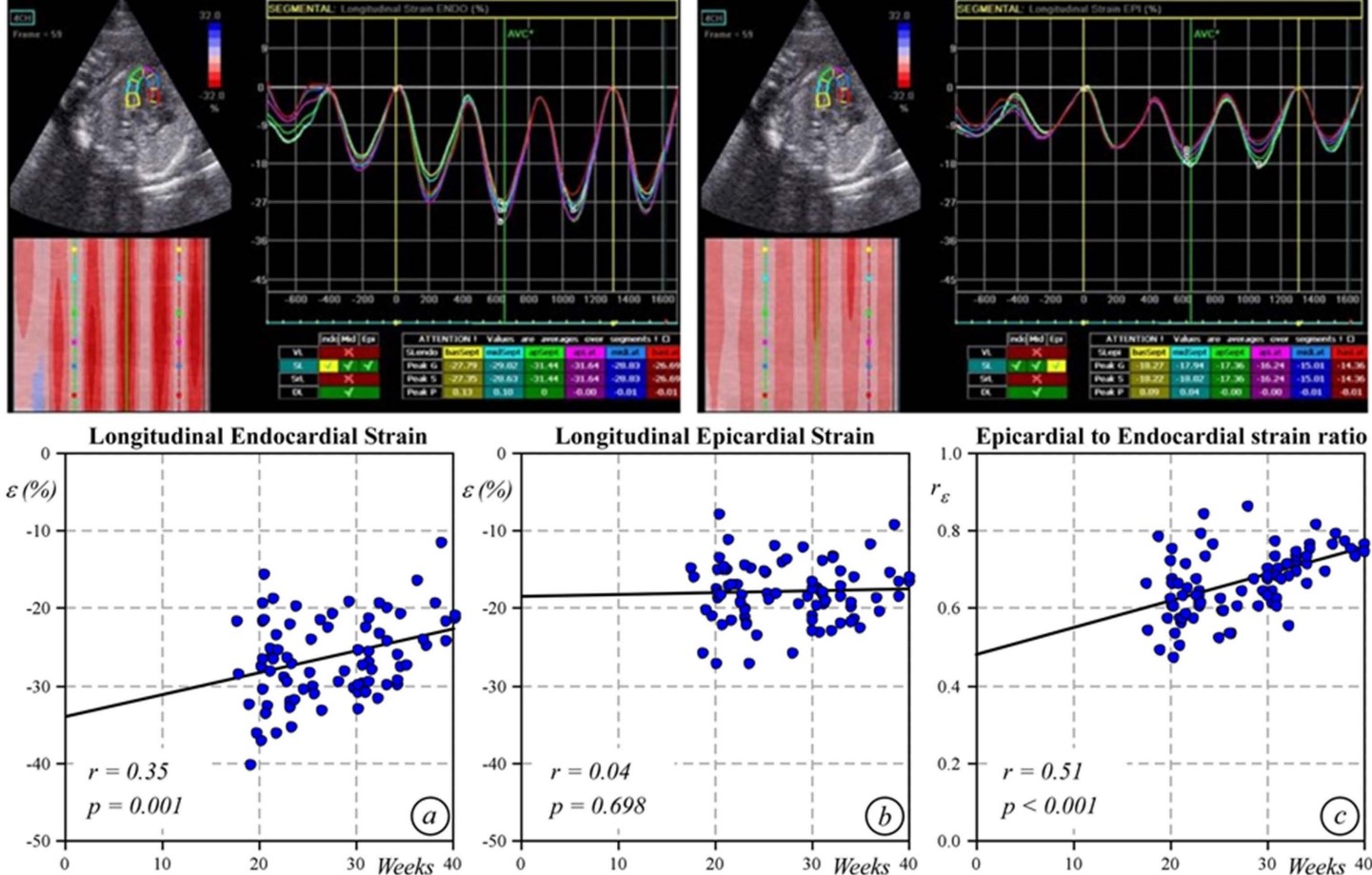

**Fig 3. Speckle tracking echocardiography. On the top, left ventricular strain curves. Bottom, plots of endocardial longitudinal strain values (a), epicardial longitudinal strain values (b), and epicardial to endocardial longitudinal strain ratio (c) at different gestational ages.**

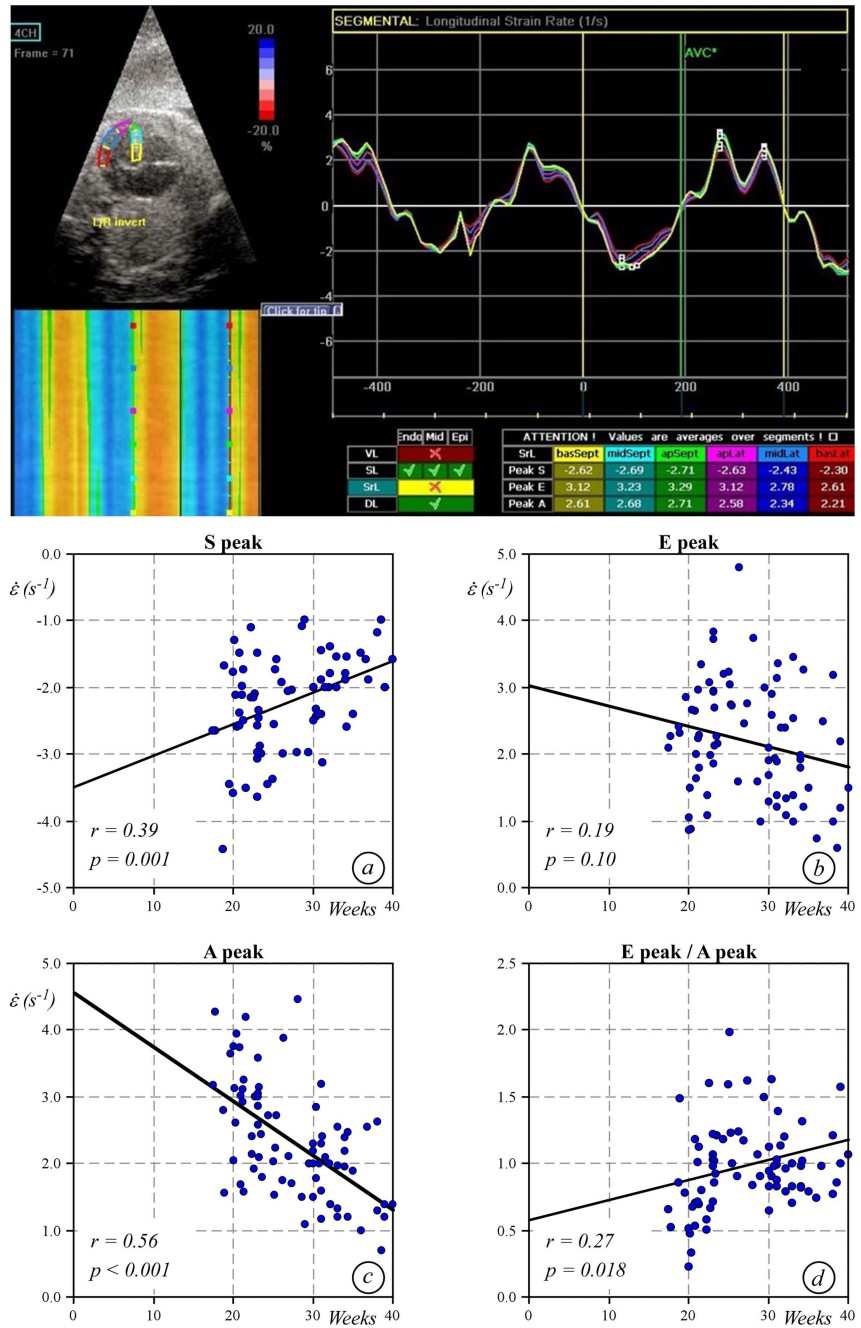

**Fig 4. On the top, strain rate curves with S, E and A peaks. Bottom, strain rate plots at different gestational ages: systolic peak (a), early diastolic peak (b), active atrial contraction peak (c) and early diastolic to atrial contraction peak ratio (d).**

peak by strain rate decreased with gestational age (r=0.385; p=0.001), whereas the E remained quite stable (r=-0.19; p=0.10) and the A wave progressively decreased (r=- 0.56; p<0.001) with gestational age. The E/A ratio increased over time (r=0.27; p=0.018). As expected, wall thickness and diameters grew following a linear regression model, while the left ventricular volume increased based on a cubic regression model (Fig 5).

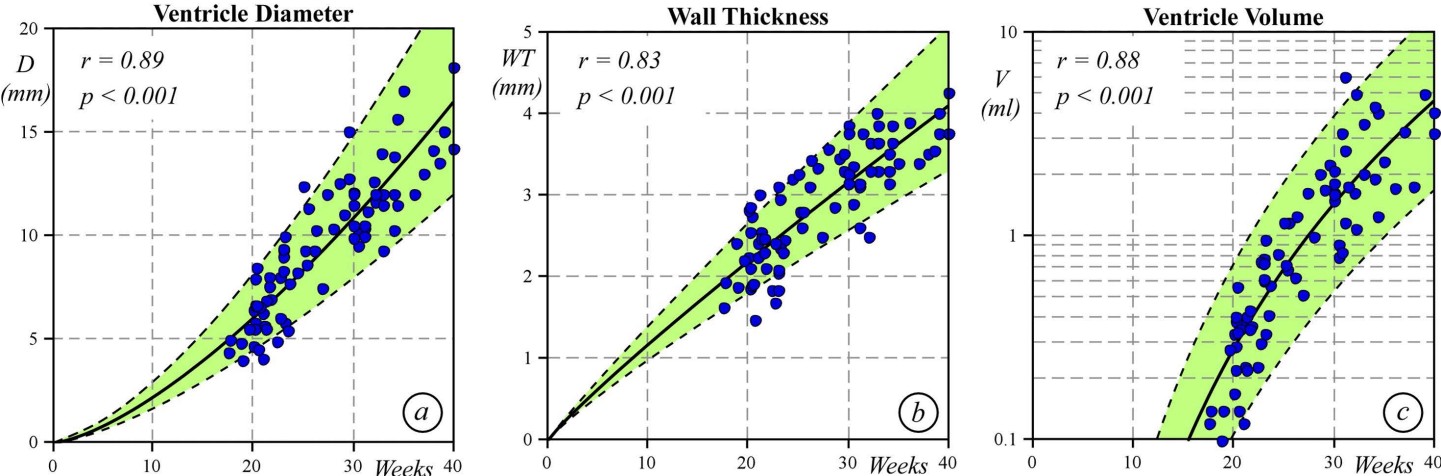

**Fig 5. Left ventricular end diastolic diameter (a), left ventricular wall thickness (as mean of basal septum and basal lateral wall end diastolic thickness) (b), and left ventricular end diastolic volume (c) [in logarithmic scale] at different gestational ages.**

## Histology

The thickness of the left ventricular wall increased with gestational age, from 1305 µm at 13 weeks of gestation to 5782 µm at 40 weeks of gestation. The relative proportion of the endocardial layer remained stable, representing around the 50% of the total myocardial thickness (r=0.058; p=0.8). In contrast, the proportion of epicardial layer increased four-fold, representing 7% of the fetal myocardium at 13 weeks and increasing to 24% by 37 weeks (20 weeks 17%, 30 weeks 22%, 40 weeks 30%), with a significant increase epicardial to endocardial wall thickness with gestational age (r=0.76; p<0.0001). Finally, the relative thickness of the mesocardium decreased with gestation (r=0.51; p=0.02). Like speckle-tracking echocardiography data on epicardial to endocardial strain ratio, the histological relative epicardial to endocardial wall thickness ratio progressively increased with gestational age (Figs 6 and 7) (r=0.76; p<0.0001; y=0.011x+0.05). We found an excellent correlation between the epicardial to endocardial strain ratio and epicardial to endocardial wall thickness (r=0.950, p<0.001).

The angle of the fibers progressively decreased from 90 to 0 from the endocardium to the mesocardium and from 0 to -90° from the mesocardium to the epicardium. We divided myocardium in seven layers, in Fig 8 was shown the mean angle for the single layer. The angle gradient was stable in the endocardial layer, while was steeper in the epicardium at 13 weeks of gestation, reducing the angle gradient proportionally with the thickening of the epicardial layer.

## Discussion

### Normal fetal myocardial development & function

The heart develops from a tubular structure, which by a process of looping and torsion, evolves in a four-chamber structure [17]. The macroscopic anatomy of the heart is almost complete at eight weeks of gestation, whereas the ultrastructure continues to evolve up to birth and beyond. In fact, all the fibers of the embryological heart are arranged radially, whereas the adult heart is highly anisotropic [14,18]. Besides being arranged in two layers, the fibers undergo a toroidal rotation within the layers [19]. Thus, the inner fibers are more longitudinal, becoming transverse in the middle part to return progressively more longitudinal

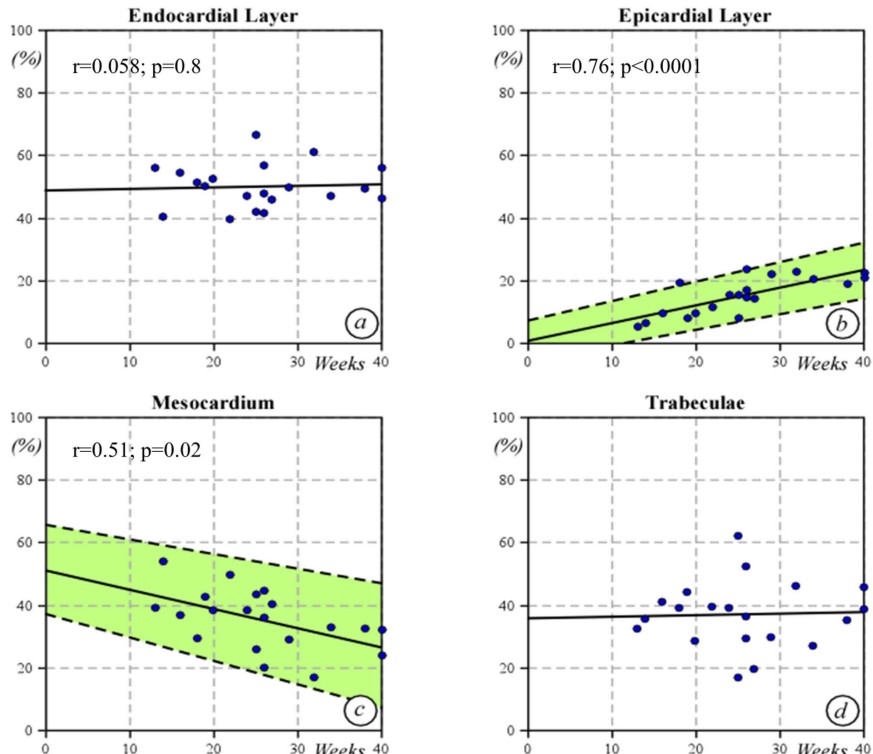

**Fig 6. Relative wall thickness of endocardial (a), epicardial (b) and mesocardium (c) by histological evaluation. The proportion of trabeculated myocardium (d) remained unchanged during gestational age.**

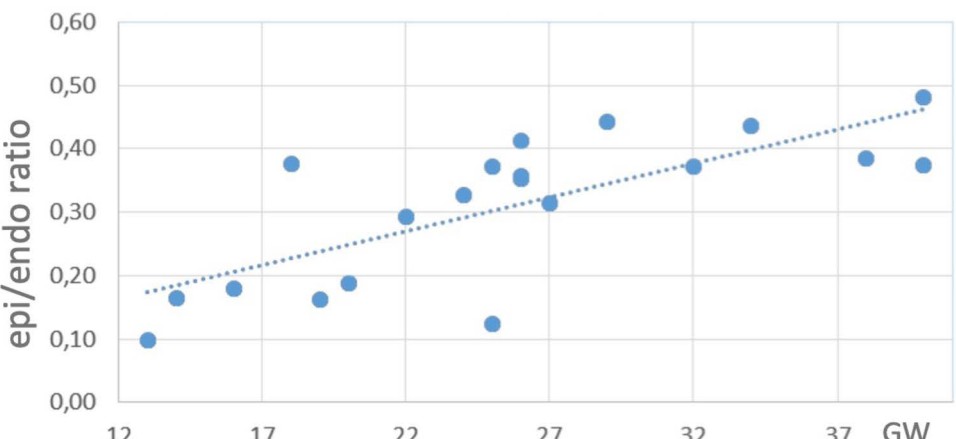

**Fig 7. Endocardial to epicardial wall thickness ratio by histology. There was a progressive increase over gestational weeks (GW).**

toward the epicardium. Using 3 Tesla MRI, Mekkaoui et al. [5] studied the myocardial tissue of 4 hearts from 10 weeks of gestation to birth. They found that myocardial fibers gradually develop from a single transverse layer to a double longitudinal layer at birth. These results agreed with our histological data.

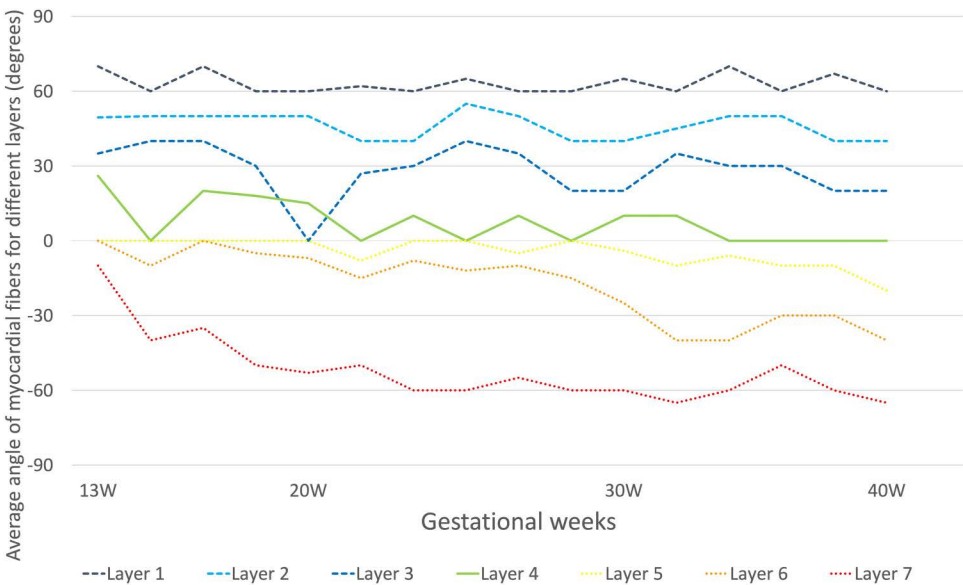

**Fig 8. Changes in fibers' angulation during gestational age based on histology. The myocardium was divided in seven layers. Layers from 1 to 3 (dashed lines) represent endocardial layers, layer 4 (continue line) is the mesocardial layer, the layers 5 to 7 (dotted lines) become epicardial layer.**

In our paper, for the first time, the layer specific longitudinal strain and the correlation between fiber orientation, histological changes and in vivo functional changes were studied in fetal hearts. The data found confirm that the endocardial longitudinal layer develops first and is well represented since the first evaluation at 17 GW, while the epicardial longitudinal layer maturation is slower and completed just at the end of the gestation. These histological changes, associated to extracellular matrix development and to the cardiomyocyte maturation [20,21], might suggest a progressive improvement on systolic left ventricular function, as demonstrated in studies on preterm infants [6–7]. The transition from the fetal circulation to the postnatal circulation generates deep changes in the left ventricular function. James et al. [6] demonstrated a progressive improvement of left ventricular twist in the first week of life in infants less than 29 weeks of gestation, confirming the fast changes in left ventricular mechanics driven by the dramatic changes in the newborn heart function. Our previous study demonstrated that the mechanism driving the changes in the left ventricular rotation and twist in preterm newborns was the improvement of the epicardial longitudinal strain [7], according to the delayed epicardial development during fetal life. These findings may be at least in part responsible of the lower tolerance to pre-load and after-load variation in preterm newborns [6,22].

Despite several studies were addressed to estimate normal longitudinal strain values during pregnancy, there are still conflicting data on the physiologic myocardial longitudinal function development and normal range values [23,24]. These differences may be explained with different study design, different machine used and different software type and version available [24,25]. In particular, a high frame rate (>60–80 fps) should be obtained to have an adequate number of frames for heart beat. Thus, a dedicated protocol of image acquisition is mandatory for this kind of studies. In addition, in the earliest versions, the software for speckle tracking analysis were designed and validated for adult left ventricular function assessment [11]. This means that their ideal range of myocardial wall thickness was between 4 and 12 mm. A too large region of interest can be the reason of too low longitudinal strain values at 20–24 weeks

found in earliest studies addressed on the normal strain values during fetal growth. Third, the lack of ECG gating requires a manual adjustment of the heart cycle and exclude the possibility to analyze simultaneously more than one heart cycle. The extensive "off-label" use of speckle tracking echocardiography for the assessment of right ventricle and atria function promoted variations and improvement on the software so that the wall thickness can be now manually adjusted and tailored on the single segment and not just pre-fixed by the software. In conclusion, in expert hands, the software available are now ready for fetal heart speckle tracking analysis. In addition, our study for the first time used a layer-specific approach to study the fetal heart and validated the capability of speckle tracking software to distinguish between longitudinal endocardial and epicardial deformation in fetal hearts with a histological study focused on the fiber orientation and the layer thickness of the left ventricular wall. Our data showed that the sub-endocardial strain decreases from the 17th to the 40th week of gestation (from -27.5% to -24.8%). On the other hand, the sub epicardial longitudinal strain increases from -17.1% to -18.6%.

### Relevance of findings to clinical pathology

Speckle tracking echocardiography was recently used to provide new insights into the understanding of fetal heart function in specific disease states. Miranda et al [26] used speckle tracking to study fetuses of mother with diabetes, demonstrating a biventricular diastolic dysfunction and right ventricular systolic dysfunction by deformation analysis in the third trimester of pregnancy. DeVore et al. [27] proposed a multi-parametric approach, including longitudinal strain, to study the left and right contractility in fetuses with intrauterine growth restriction. They concluded that high rates of abnormal ventricular contractility were present in fetuses with an estimated fetal weight <10th centile, irrespective of the Doppler findings of the pulsatility index of the umbilical artery, and/or cerebroplacental ratio [27].

Speckle tracking echocardiography was used to predict the post-natal outcome of congenital heart diseases. Cohen et al. [28] found that fetuses with pulmonary atresia and intact ventricular septum (PAIVS) had a reduced left ventricle global longitudinal strain value compared to normal, and the patient with post-natal evidence of right ventricle dependent coronary circulation had the worst values compared to PAIVS patient without sinusoids (-15.8±1.2% vs -17.9±1.7%, p=0.009; normal -23.7±2.0%).

Wohlmuth et al [29] studied the outcome of fetal balloon aortic valvuloplasty with Tissue Doppler (TDI). They found that the fetuses that improved TDI data (S', E', E'/A', E/E' and TDI-based myocardial performance index) had a post-natal biventricular physiology outcome. On the other hand, no changes in TDI data predicted a single ventricle physiology correction.

In congenital heart disease like PAIVS and aortic stenosis, the sub-endocardial ischemia and fibroelastosis play a central role in the loss in function of the left ventricle [30]. Fibrous tissue has isotropic elastic properties while myocardial fibers are longitudinally orthotropic and transversally isotropic [31]. Myocardial fibroelastosis replaces the endocardial longitudinal layer and, by increasing the stiffness of the endocardial layer. This process results in a restrictive pattern with high end-diastolic pressures. Based on these data, the endocardial longitudinal strain ratio could predict the amount of endocardial fibroelastosis and/or the response to fetal balloon valvuloplasty, improving the patient stratification for this complex and risky procedure [32].

### Limitations

One limitation of our study is that the anatomical and functional data were not obtained from the same patients. We collected in vivo echocardiographic data from cardiovascular healthy

fetuses between 17 and 36 weeks of gestation and we compared with post-mortem data of fetuses (without cardiovascular abnormalities and with normal growth before the death) between 13–40 weeks of gestation. Although only well-preserved fetal hearts were selected for histological study, we were unable to evaluate the angle of myofibers due to tissue maceration in 4 cases. Furthermore, the sample size was too small to develop myocardial fiber angulation plots for gestational age. Fig 8 simply represents the value obtained from the single sample at a given gestational age.

Recently, Semmler et al demonstrated angle dependence of speckle tracking in a fetal population by using a Vitrea Software (Canon® Medical System) [33]. However, in our study we selected only vertically oriented apical 4 chamber view with a good or excellent acoustic window. We specifically avoided the analysis of transverse apical 4 chamber view. Finally, this study for the first time used layer specific longitudinal strain on fetal hearts. Thus, the standard deviation was higher than values calculated in full-thickness strain values. However, the image acquisition and post-processing protocol was already validated on small animals and rodent [34] and data obtained were validated by histologic data.

## Conclusions

Left ventricular myocardium maturation begins early during fetal life. As the fetus develops, both the relative tissue volume and peak systolic strain rates shift together from the endocardium towards the epicardium. It is a slow process, completed late in fetal life.

## Acknowledgements

We thank Jean Ann Gilder (Scientific Communication srl., Naples, Italy) for text editing.

## Author contributions

**Conceptualization:** Biagio Castaldi, Daniela P. Boso, Francesca M. Susin, Ornella Milanesi, Annalisa Angelini.

**Data curation:** Biagio Castaldi, Marny Fedrigo, Irene Cattapan, Elena De Filippi, Paolo Peruzzo, Giulia Comunale, Paola Veronese, Annalisa Angelini.

**Formal analysis:** Biagio Castaldi, Marny Fedrigo, Daniela P. Boso, Elena De Filippi, Paolo Peruzzo.

**Investigation:** Biagio Castaldi, Marny Fedrigo, Irene Cattapan, Giulia Comunale.

**Methodology:** Biagio Castaldi, Giovanni Di Salvo.

**Project administration:** Biagio Castaldi.

**Software:** Irene Cattapan.

**Supervision:** Giovanni Di Salvo.

**Validation:** Francesca M. Susin.

**Writing – original draft:** Biagio Castaldi, Paolo Peruzzo.

**Writing – review & editing:** Daniela P. Boso, Francesca M. Susin, Ornella Milanesi, Annalisa Angelini, Giovanni Di Salvo.

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
