## [Decision Letter · Decision Letter 0]

12 May 2022

PONE-D-22-00611Development of the Fetal Myocardium: The Impact of Fiber Orientation on Left Ventricular MechanicsPLOS ONE

Dear Dr. Castaldi,

Thank you for submitting your manuscript to PLOS ONE. After careful consideration, we feel that it has merit but does not fully meet PLOS ONE’s publication criteria as it currently stands. Therefore, we invite you to submit a revised version of the manuscript that addresses the points raised during the review process.

 Your paper has been reviewed by two experts in the field of heart development. Though the paper was considered interesting, concerns were expressed regarding several technical aspects of your study

that call into question the conclusions of the paper. In particular, concerns were raised regarding the reliability of the STE measures, the lack of other (additional) measures of maturation and the usefulness of the simulations

in relationship to the experimental measures as well as the histological measurements. It was suggested that the author attempt to validate their findings in an animal model and consider additional factors in the faetal demise.

The Editors concur with these comments and invite the authors to address comments made

We look forward to receiving your revised manuscript.

Kind regards,

Peter H. Backx

Academic Editor

PLOS ONE

Journal Requirements:

3. You indicated that you had ethical approval for your study. In your Methods section, please ensure you have also stated whether you obtained consent from parents or guardians of the minors included in the study or whether the research ethics committee or IRB specifically waived the need for their consent.

Reviewers' comments:

Reviewer's Responses to Questions

**Comments to the Author**

1. Is the manuscript technically sound, and do the data support the conclusions?

Reviewer #1: No

Reviewer #2: Yes

2. Has the statistical analysis been performed appropriately and rigorously? 

Reviewer #1: I Don't Know

Reviewer #2: Yes

3. Have the authors made all data underlying the findings in their manuscript fully available?

Reviewer #1: No

Reviewer #2: Yes

4. Is the manuscript presented in an intelligible fashion and written in standard English?

Reviewer #1: Yes

Reviewer #2: Yes

5. Review Comments to the Author

Reviewer #1: The authors formulated the following 3 Aims:

1. "to assess in vivo the endocardial and epicardial longitudinal strain in fetuses of different gestational ages by STE as surrogate of layer maturation"

2. to confirm, on histological sections, the relative wall thickness of the endocardial and epicardial layers at different gestational ages.

3. to evaluate in silico the effect of different myocardial fiber arrangements on diastolic performance.

The endocardial and epicardial strain measurements using STE by their nature are very noisy. During early phases of gestation, when the heart is small the level of noise should be even higher. It is not clear what was the point of using strain as a measure of maturation given there are more common and more reliable measures of maturation such as ventricle volume, diameter or wall thickness. Figure 2 and Table 1 contain overlapping information. Table 1 is not necessary and should be omitted.

It is not clear what histological studies were intended to confirm. The criteria used to identified the boundaries between different layers are not described. Neither is clear how the fiber angle was measured and quantified in particular in situations illustrated in Figure 1,b3.

Simulations utilize a 7-layer mechanical model, which is loosely connected to the rest of the study. It simulates pressure-volume curves which were not measured in this study. Reference 37 cited in the legend to figure 7 is not present in the list of references. It is not clear how the experimental values of fiber angle were determined. According to the methods sections histological analysis involved only 4 layers.

Overall the study is poorly designed and conclusions are not supported by the experimental information.

Reviewer #2: This is an interesting paper that seeks to characterize the fiber orientation in the fetal myocardium and understand its impact on LV mechanics.

1. There are small grammatical issues throughout paper and abstract that should be addressed.

2. Use of histological sections from fetal hearts. A portion of these were from fetal demise- do you know cause? . While structurally normal heart development, issues with placenta and vascularization can affect myocardial development prior to fetal demise. This may affect some of your models.

3. limitations related to angle dependence acknowledge. Are you able to use data from well characterized animal models with histology to validate the in silico model

6. PLOS authors have the option to publish the peer review history of their article (what does this mean? ). If published, this will include your full peer review and any attached files.

**Do you want your identity to be public for this peer review?** For information about this choice, including consent withdrawal, please see our Privacy Policy .

Reviewer #1: No

Reviewer #2: No

---

## [Author Response · Author response to Decision Letter 1]

8 Jul 2022

Reviewer #1:

The authors formulated the following 3 Aims:

1. "to assess in vivo the endocardial and epicardial longitudinal strain in fetuses of different gestational ages by STE as surrogate of layer maturation"

2. to confirm, on histological sections, the relative wall thickness of the endocardial and epicardial layers at different gestational ages.

3. to evaluate in silico the effect of different myocardial fiber arrangements on diastolic performance.

The endocardial and epicardial strain measurements using STE by their nature are very noisy. During early phases of gestation, when the heart is small the level of noise should be even higher. It is not clear what was the point of using strain as a measure of maturation given there are more common and more reliable measures of maturation such as ventricle volume, diameter or wall thickness.

Thank you for your comment. STE suffers of some inter- and intra- observer variability. However, it is comparable/lower to other echocardiographic parameters. There are several papers demonstrating the feasibility of STE in fetal hearts, since 16 WoG. However, we stated some issues in the limitations (page 15, lines 430-436). On the other hand, strain imaging still remains an effective tool to study in vivo and not-invasively myocardial function. We reported common measures of heart maturation (page 9 lines 242-244 and figure 4), however, these parameters do not allow to study the fibers distribution. On the other hand, STE is a validated method to estimate fiber arrangement in the heart.

Figure 2 and Table 1 contain overlapping information.

Table 1 is not necessary and should be omitted.

Sorry, I didn’t understand. Figure 2 indicates longitudinal strain values by echo; table 1 is the fibers’ angles by histological measures.

It is not clear what histological studies were intended to confirm. The criteria used to identify the boundaries between different layers are not described. Neither is clear how the fiber angle was measured and quantified in particular in situations illustrated in Figure 1,b3.

We would like to thank the reviewer for raising the point related to the methodological aspects of histology which were not reported in the first version of the paper. We have specified how we proceeded for quantification and identification of the fiber orientations and angle evaluation in the material and methods session on page 7 and 8 on line 175-209. We added two reference on this topic, too.

We believe that the histological study was important for supporting the in silico modelling of different layers orientation to characterize the fiber orientation in the fetal myocardium and understand its impact on LV mechanics.

Simulations utilize a 7-layer mechanical model, which is loosely connected to the rest of the study. It simulates pressure-volume curves which were not measured in this study. Reference 37 cited in the legend to figure 7 is not present in the list of references. It is not clear how the experimental values of fiber angle were determined. According to the methods sections histological analysis involved only 4 layers.

Sorry for the mistake, we added ref. 37 (now 39) to the bibliography. The 7 layers model was used to create different models by changing fibers angulation. According to our and previous studies, the fiber angulation changes progressively from the endocardium to the epicardium. A 7 layers model was considered the most appropriate compromise between simplification of the model to allow data elaboration and adherence to in-vivo model. The angle of the single layer was attributed according to those found by histological study at the given gestational age. The models were tested by comparing the physiologic pressure-volume plots to that generated by the computational model in order to verify the goodness to fit. Figure 7 demonstrate the adequate adherence of the model to the physiological pressure-volume plot found in animal models and in neonates from previous studies.

In figure 8, we started from the 40GW arrangement and then we changed the fiber orientation to verify alternative setting and the role of alternative fibers arrangements on the pressure-volume plot.

Overall the study is poorly designed and conclusions are not supported by the experimental information.

Thank you for your comment. This is the first study facing the fetal development of myocardium in humans. To build an animal model in this setting is very difficult, and to obtain invasive data on healthy fetal heart is impossible, so we did the best (based on our knowledge) to confirm in vivo data with histological and computational models. Computational models are a validated alternative to animal model, at least to build an adequate hypothesis before to plan further and more invasive or ethical challenging studies.

We removed the last sentence of the conclusion, that might sound a little beat speculative.

Reviewer #2:

This is an interesting paper that seeks to characterize the fiber orientation in the fetal myocardium and understand its impact on LV mechanics.

1. There are small grammatical issues throughout paper and abstract that should be addressed.

Thank you for this comment. We reviewed the English language.

2. Use of histological sections from fetal hearts. A portion of these were from fetal demise- do you know cause? While structurally normal heart development, issues with placenta and vascularization can affect myocardial development prior to fetal demise. This may affect some of your models.

We would like to thank the reviewer for raising these important aspects, which we had considered in the selection of our cases but we did not specify in the first version. We excluded cases with intrauterine growth restriction and cases with maternal malperfusion vascularization. Moreover, according to fetal anthropometric indexes the selected hearts were within a normal range for the gestational weeks. We have specified it in the material and method session on page 6 lines 144-146.

As for the cause of intrauterine death of our cases: 4 were unknown, 1 for volvulus, 1 for uterine rupture, 3 for cordonal issues, 2 for infection; among the miscarriage 2 for cervical incontinence, 1 for early placenta abruption, 1 for superficial placental implantation.

3. limitations related to angle dependence acknowledge. Are you able to use data from well characterized animal models with histology to validate the in silico model.

Thank you for this suggestion. We agree that an animal model might improve the validation of in-silico model. Unfortunately, to build a fetal animal model is outside the purpose of this study, also because a very special ethical authorization should be needed to study fetuses of superior mammalians. In addition, before to build an animal model, a clear hypothesis should be presented to the ethical committee, so a pilot study is mandatory, and this study works in this direction.

Finally, the development of a similar animal study is too long to add those data to this paper.

---

## [Decision Letter · Decision Letter 1]

5 Feb 2023

PONE-D-22-00611R1Development of the Fetal Myocardium: The Impact of Fiber Orientation on Left Ventricular MechanicsPLOS ONE

Dear Dr. Castaldi,

Thank you for submitting your manuscript to PLOS ONE. After careful consideration, we feel that it has merit but does not fully meet PLOS ONE’s publication criteria as it currently stands. Therefore, we invite you to submit a revised version of the manuscript that addresses the points raised during the review process.

Your paper has been reviewed again.  Only one reviewer previously reviewed you manuscript.  I apologize for the long period taken to get your paper reviewed.

 As you can see from the reviews, both reviewers are asking for major changers to the paper.  If you wish to revise the manuscript, the following changes must be made before the paper will be accepted:

1) "Reframing the scientific question and using the in silico model as the central focus" is recommended in order to address the many limitations and assumptions made in analyzing and interpreting the histological results.

2.  Providing additional clear details on the histological analyses and results (addressing questions such as: a)  how the various layers identified? and b) what were the thicknesses of the various layers?

3)  Clarification on how the histological data was incorporated into the in silico calculations and a discussion of the relationship between the histological measures and in silico results.

We look forward to receiving your revised manuscript.

Kind regards,

Peter H. Backx

Academic Editor

PLOS ONE

Additional Editor Comments:

Your paper has been reviewed again. Only one reviewer previously reviewed you manuscript. I apologize for the long period taken to get your paper reviewed.

As you can see from the reviews, both reviewers are asking for major changers to the paper. If you wish to revise the manuscript, the following changes must be made before the

paper will be accepted:

1) "Reframing the scientific question and using the in silico model as the central focus" is recommended in order to address the many limitations and assumptions made in analyzing and interpreting the

histological results.

2. Providing additional clear details on the histological analyses and results (addressing questions such as: a) how the various layers identified? and b) what were the thicknesses of the various layers?

3) Clarification on how the histological data was incorporated into the in silico calculations and a discussion of teh relationship between the histological measures and in silico results.

Reviewers' comments:

Reviewer's Responses to Questions

**Comments to the Author**

1. If the authors have adequately addressed your comments raised in a previous round of review and you feel that this manuscript is now acceptable for publication, you may indicate that here to bypass the “Comments to the Author” section, enter your conflict of interest statement in the “Confidential to Editor” section, and submit your "Accept" recommendation.

Reviewer #2: (No Response)

Reviewer #3: (No Response)

2. Is the manuscript technically sound, and do the data support the conclusions?

Reviewer #2: Partly

Reviewer #3: Partly

3. Has the statistical analysis been performed appropriately and rigorously? 

Reviewer #2: N/A

Reviewer #3: I Don't Know

4. Have the authors made all data underlying the findings in their manuscript fully available?

Reviewer #2: Yes

Reviewer #3: No

5. Is the manuscript presented in an intelligible fashion and written in standard English?

Reviewer #2: Yes

Reviewer #3: No

6. Review Comments to the Author

Reviewer #2: The authors have resubmitted their manuscript with changes in grammar and clarification of some of the reviewers comments.

The use of STE , histology and in silico models have been used to try to approximate fiber orientation over fetal development

Limitations of the study exist and many assumptions and approximations are made given the issues obtaining fetal tissue and testing hypothesis.

As the study stands, it is not fully supported by its scientific design. Reframing the scientific question and using the in silico model as the central focus and using either animal and/ or the human data as potential supporting evidence may help strengthening the paper

Reviewer #3: The authors attempt to assess the histological and physiological maturation steps of the LV myocardium during the fetal development. In-silico calculations are also performed.

Measurements/Calculations:

1) longitudinal endocardial and epicardial strain by echocardiography in 105 fetuses.

2) layer thickness and cardiac fiber orientation in 20 “non-diseased” fetal hearts.

3) in-silico models of the left ventricular myocardium corresponding

Results: Progressive increases in epicardial/endocardial longitudinal strain ratio were seen with gestational age. Histological data revealed proportional increases in the epicardial layer during development. In silico calculations showed compliance reduction as the longitudinal fibers were more vertical.

Conclusion: “Left ventricular myocardium maturation begins early during fetal life and starts from the differentiation of a subendocardial layer. The development of the epicardial layer is slower and it is completed late in fetal life.”

Assessment

This is revised manuscript. I did not review the paper in the first round. I believe that the major concerns raised in the previously reviews were only slightly addressed. The remaining concerns include the following:

1) Details of the histological analyses and results remain missing, as pointed out by the previous reviewer 1. Were the various layers identified? How were they identified? What were the thicknesses of the various layers?

2) How (and what part of) the histological data was incorporated into the in silico calculations is not described or discussed. It seems the in silico calculation are performed without any specific regards for histological results (i.e. fiber orientations). Would it not be critical to incorporate these histological measures and see how this impacts on calculated function? I would expect that these calculations would require some assumptions about tissue compliance in the fiber direction but there is no description of this.

7. PLOS authors have the option to publish the peer review history of their article (what does this mean? ). If published, this will include your full peer review and any attached files.

**Do you want your identity to be public for this peer review?** For information about this choice, including consent withdrawal, please see our Privacy Policy .

Reviewer #2: No

Reviewer #3: No

---

## [Author Response · Author response to Decision Letter 2]

1 Jul 2023

Reviewer #3 comments:

• This is revised manuscript. I did not review the paper in the first round. I believe that the major concerns raised in the previously reviews were only slightly addressed. The remaining concerns include the following:

o 1) Details of the histological analyses and results remain missing, as pointed out by the previous reviewer 1. Were the various layers identified? How were they identified? What were the thicknesses of the various layers?

Thank you for your comment. We described more in detail in the Methods (lines from 152 to 186 – tracked changes file). In particular, we described how we computed the thickness of the layer and how we calculated the angle. The layers were distinguished according to the different orientation of the fibers, causing a different grade of orange (lighter in longitudinal fibers). In addition, we measured the angle of the fibers in the different layers. We specified that the myocardium was divided in seven layers to calculate the different angle of the fibers, similarly to Streeter et al. We divided the myocardium in 7 layers to have an adequate sample size for repeated measures of the fibers’ angle and to match the results found from histology with the in-silico model. In the results, we reported the values of wall thickness (from 1305 �m at 13 weeks of gestation to 5782 �m at 40 weeks of gestation) and the changes in fibers’ angulation in the seven layers (figure 6). The relative layer thickness was reported in Figure 5.

o 2) How (and what part of) the histological data was incorporated into the in silico calculations is not described or discussed. It seems the in silico calculation are performed without any specific regards for histological results (i.e. fiber orientations). Would it not be critical to incorporate these histological measures and see how this impacts on calculated function? I would expect that these calculations would require some assumptions about tissue compliance in the fiber direction but there is no description of this.

Thank you for this question. We specified that in silico model was based on the histological data, in particular on the fibers’ orientation found in the different layers. Thus, the mesocardial thickness changed over the time, as well as the epicarial layer. On the other hand, endocardial layer remained unchanged because the relative wall thickness of that layer remained stable since the 13th GW. We built a 7 layers model to merge at the best of our possibilities histological data and computational methods. In addition, data on LV volumes at different gestational ages were acquired from fetal echo. Data on elastic properties of myocardial fibers, and shear stress and pressure-volume plots data were available from previous studies (ref. 17-19 and 38-39) on humans or from animal fetuses. For ethical reasons, the acquisition of these data was not feasible in human fetuses. We already stated this point in the limitations (lines 379-384 tracked changed version). However, we believe that these data were adequate for our model, and the pressure-volume plots obtained from our model was in agreement with normal physiology of the LV.

---

## [Decision Letter · Decision Letter 2]

12 Jul 2023

PONE-D-22-00611R2Development of the Fetal Myocardium: The Impact of Fiber Orientation on Left Ventricular MechanicsPLOS ONE

Dear Dr. Castaldi,

Thank you for submitting your manuscript to PLOS ONE. After careful consideration, we feel that it has merit but does not fully meet PLOS ONE’s publication criteria as it currently stands. Therefore, we invite you to submit a revised version of the manuscript that addresses the points raised during the review process.

We look forward to receiving your revised manuscript.

Kind regards,

Peter H. Backx

Academic Editor

PLOS ONE

Additional Editor Comments:

Your paper has been reviewed by one of the previous Reviewers. The paper continues to have major deficiencies. In the 2 previous review cycles, the Reviewers were highly critical of the in-silico results.

Please address completely the comments made by the attached Review. Without fully addressing all the Reviewers comments, your paper will be be deemed inadequate for publication.

Reviewers' comments:

Reviewer's Responses to Questions

**Comments to the Author**

1. If the authors have adequately addressed your comments raised in a previous round of review and you feel that this manuscript is now acceptable for publication, you may indicate that here to bypass the “Comments to the Author” section, enter your conflict of interest statement in the “Confidential to Editor” section, and submit your "Accept" recommendation.

Reviewer #3: (No Response)

2. Is the manuscript technically sound, and do the data support the conclusions?

Reviewer #3: Partly

3. Has the statistical analysis been performed appropriately and rigorously? 

Reviewer #3: I Don't Know

4. Have the authors made all data underlying the findings in their manuscript fully available?

Reviewer #3: No

5. Is the manuscript presented in an intelligible fashion and written in standard English?

Reviewer #3: No

6. Review Comments to the Author

Reviewer #3: The manuscript has been improved marginally in revision.

I have taken the time to provide extensive and specific guidance for revising the paper.

As stated by previous reviewers, the value and details of the in silico resuts need to be better presented. If the authors are unable to do this, then I recommend removing all these results.

The following changes need to be made carefully and completely to the manuscript.

1. In the Abstract, there are numerous punctuation errors and run-on sentences. Please correct. Also you have included descriptions of the methods in the section dedicated to Aims of the studies. Please rearrange. The Abstract uses abbreviation that are not appropriately defined (what is STE and GW, for example).

2. What is meant by “Using the function “measurements” ”? Why are you uses quotes. Do you mean the data from the echo measurements? If so, just state this.

3. It would be useful to accurately define the “mean S, E and A peak values and the E/A ratio. Show a typical epicardial and endocardial Doppler signal for a typical heart that was used to assess the S waves. Place this in Figure 2.

4. The authors need to state how they were able to align the histology measurements with the echo measurements. Some details on this was achieved using the “software” is needed. What assumptions were made? What is the resolution of the echo in comparison with the histology. I am not convinced this possible.

5. Table 1 has an incorrect title. You state “Muscle fiber orientation in the left ventricle layers of a fetus at three gestational ages based on hystological data.” Hystological is incorrectly spelled. It is histological. The angle are not based on specific data; it is based on assumptions. You are better to something like “The angles were assumed and are overall consistent with the histological measurements”.

6. Put statistics into Table 2 and also include the correlation data.

7. What is being shown in Figure 6? Is this based on Echo data? How can such changes “in fibers’ angulation during gestational age” be measured with any level of certainty using echo? If this data comes from echo measurements, I have significant doubts that these results are meaningful. Please explain, justify and defend.

8. Figure 7 is of limited value and needs to be removed. It lacks dimension etc.

9. The in silico model is difficult to understand without further explanation. Are the pressures plotted in Figure 8, systolic or diastolic? Since the histology measurements are all with hearts in diastole, it would only make sense to explore the results of an in silico model for diastolic pressures and volumes. Yet, the description of the results does make this clear.

10. The authors state “As shown in Figure 8d and 8e, the shape of the left ventricle was deeply unrealistic, and compliance of radially oriented fibers was much lower.” Which result? Both the 0O and 90O cases? It is not at all obvious that the volumes shown in Figure 8 a-c are realistic. For a 40GW heart, the volumes for diastolic volumes seem to be unexpectedly small. Do experimental data exist in order to compare to the results of Figure 8?

11. I see little point in illustrating the data in Figure 9. What is the purpose of these results? In particular, the authors state “As shown in Figure 9 (dashed lines), the early development of the endocardial layer resulted in a pressure volume plot similar to the

definitive array”. What is the definitive array? The meaning of this statement is unclear. The authors then state “the inverse plot resulted in poor compliance.” Again, what is the basis for such a statement? Finally, the authors state “The comparison between the 90°/0° and 0°/90° cases showed that the inner layer plays the major role in ventricle stiffness.” Once again, the basis for such a conclusion is a mystery.

In summary, unless the authors are willing to present their data in a careful, coherent and thoughtful manner, and unless the authors are better able to justify their insilico model with additional details in describing the result, I cannot be supportive of publication. I note that this paper has been reviewed twice previously and my comments align well with previous critiques.

7. PLOS authors have the option to publish the peer review history of their article (what does this mean? ). If published, this will include your full peer review and any attached files.

**Do you want your identity to be public for this peer review?** For information about this choice, including consent withdrawal, please see our Privacy Policy .

Reviewer #3: No

---

## [Author Response · Author response to Decision Letter 3]

17 Aug 2023

Reviewer #3: The manuscript has been improved marginally in revision.

I have taken the time to provide extensive and specific guidance for revising the paper.

As stated by previous reviewers, the value and details of the in silico resuts need to be better presented. If the authors are unable to do this, then I recommend removing all these results.

The following changes need to be made carefully and completely to the manuscript.

1. In the Abstract, there are numerous punctuation errors and run-on sentences. Please correct. Also you have included descriptions of the methods in the section dedicated to Aims of the studies. Please rearrange. The Abstract uses abbreviation that are not appropriately defined (what is STE and GW, for example).

Thank you. We deeply revised the abstract. In particular, we rebuilt the background and aims section, we removed the abbreviations (GW and LV) and we checked orthography.

2. What is meant by “Using the function “measurements” ”? Why are you uses quotes. Do you mean the data from the echo measurements? If so, just state this.

Thank you for this comment. It refers (line 171, page 7) to histology section. So the measure was referred to specimens. We changed the sentence in: “Using the same software, we…” (Image Pro Plus, cited in line 160…).

3. It would be useful to accurately define the “mean S, E and A peak values and the E/A ratio. Show a typical epicardial and endocardial Doppler signal for a typical heart that was used to assess the S waves. Place this in Figure 2.

Thank you for this suggestion. The figure was available in Supplementary Data, Figure A. Accordingly, we added the strain curves to figure 2 and strain rate curves to figure 3. To be more clear, we changed strain rate S, E and A value in strain rate S’, E’ and A’ values, most frequently used to identify myocardial velocities.

4. The authors need to state how they were able to align the histology measurements with the echo measurements. Some details on this was achieved using the “software” is needed. What assumptions were made? What is the resolution of the echo in comparison with the histology. I am not convinced this possible.

Speckle tracking echocardiography is a functional in vivo assessment. It is routinely used to study left ventricular mechanics. In pathological settings, decrease in strain values corresponds to loss in function of that myocardial fibers. In physiological fetal maturation of myocardium, the change in strain values can be due to a change in fiber arrangement. By histology, we confirmed that in vivo changes in strain values were associated to anatomical changes in fibers’orientation and relative thickness of endocardial and epicarial longitudinal layers. Thus, we can just correlate (rather than to align) in-vivo and in-silico data.

This is the reason of the sentence “Like speckle-tracking echocardiography data on epicardial to endocardial strain ratio, the histological relative epicardial to endocardial wall thickness ratio progressively increased with gestational age (Figure 5) (r=0.76; p<0.0001; y=0.011x+0.05).” (lines 236-238 tracked changes file)

Post-natal myocardial structure was already documented, while the fetal myocardial development was studied on animal models or (in humans) on few patients by 3T MRI. For the first time, we studied myocardial structure at different gestational age on humans. Our data were in agreement with models hypothesized models [Mekkaoui C, Porayette P, Jackowski MP, Kostis WJ, Dai G, Sanders S, Sosnovik DE. Diffusion MRI Tractography of the Developing Human Fetal Heart. PLoS One 2013;8.]. In our study, changes in layer specific speckle tracking data correspond to changes in relative layer thickness by histology. Of course, we agree that is impossible to compare in vivo function and histology from the same patient for ethical reasons.

5. Table 1 has an incorrect title. You state “Muscle fiber orientation in the left ventricle layers of a fetus at three gestational ages based on hystological data.” Hystological is incorrectly spelled. It is histological. The angle are not based on specific data; it is based on assumptions. You are better to something like “The angles were assumed and are overall consistent with the histological measurements”.

Thank you for your suggestions. We modified the title as follows: “Muscle fiber orientation in the left ventricle layers of a fetus at three gestational ages (20, 30 and 40 gestational weeks) used in the in-silico model. The angles were assumed and were overall consistent with the histological measurements. Positive angle: counterclockwise rotation; negative angle: clockwise rotation, zero degree: circumferential direction.”

6. Put statistics into Table 2 and also include the correlation data.

Thank you for your suggestion. Data were available in the text (lines 216-219). However, we added to the table, too.

7. What is being shown in Figure 6? Is this based on Echo data? How can such changes “in fibers’ angulation during gestational age” be measured with any level of certainty using echo? If this data comes from echo measurements, I have significant doubts that these results are meaningful. Please explain, justify and defend.

Sorry for the misunderstanding. Figure 6 was cited in the histology section (line 240). It was a specific request from the last review. Of course, these data cannot be obtained by echo. To be more clear, we modified the title as follows “Changes in fibers’ angulation during gestational age based on histology.”

8. Figure 7 is of limited value and needs to be removed. It lacks dimension etc.

We removed it, accordingly.

9. The in silico model is difficult to understand without further explanation. Are the pressures plotted in Figure 8, systolic or diastolic? Since the histology measurements are all with hearts in diastole, it would only make sense to explore the results of an in silico model for diastolic pressures and volumes. Yet, the description of the results does make this clear.

We agree with your comment. Pressures plotted in figure 8 (now figure 7) were diastolic. As specified in the aims and in the text, by in silico model we studied the diastolic function. However, we changed the first sentence of the in-silico section in “In-silico models were able to compare diastolic function at different gestational ages and with different myocardial fibers’ arrangements.”, and we modified the figure 7 (former figure 8) title in “(a), (b) and (c) diastolic pressure-volume plots at different gestational ages”

10. The authors state “As shown in Figure 8d and 8e, the shape of the left ventricle was deeply unrealistic, and compliance of radially oriented fibers was much lower.” Which result? Both the 0O and 90O cases? It is not at all obvious that the volumes shown in Figure 8 a-c are realistic. For a 40GW heart, the volumes for diastolic volumes seem to be unexpectedly small. Do experimental data exist in order to compare to the results of Figure 8?

Due to the changes described above, Figure 8 is now mentioned as Figure 7.

We modified the sentence in lines 250-251: “As shown in Figure 7d and 7e, the shape of the left ventricle was far from the normal shape in both the cases”.

Volumes used in figures 7a-c were based on echocardiographic data, as stated in the methods (lines 197-202). The values were absolute volumes and referred to fetal volumes and not to neonates. This might be the reason why the volumes seems to be unexpectedly small. Our data were in agreement with previous studies on fetal volumes nomograms [Messing B, Cohen SM, Valsky DV, Rosenak D, Hochner-Celnikier D, Savchev S, Yagel S. Fetal cardiac ventricle volumetry in the second half of gestation assessed by 4D ultrasound using STIC combined with inversion mode. Ultrasound Obstet Gynecol. 2007 Aug;30(2):142-51. doi: 10.1002/uog.4036. PMID: 17566143][ Simioni C, Nardozza LM, Araujo Júnior E, Rolo LC, Zamith M, Caetano AC, Moron AF. Heart stroke volume, cardiac output, and ejection fraction in 265 normal fetus in the second half of gestation assessed by 4D ultrasound using spatio-temporal image correlation. J Matern Fetal Neonatal Med. 2011 Sep;24(9):1159-67. doi: 10.3109/14767058.2010.545921. Epub 2011 Jan 21. PMID: 21250911.]

11. I see little point in illustrating the data in Figure 9. What is the purpose of these results? In particular, the authors state “As shown in Figure 9 (dashed lines), the early development of the endocardial layer resulted in a pressure volume plot similar to the

definitive array”. What is the definitive array? The meaning of this statement is unclear. The authors then state “the inverse plot resulted in poor compliance.” Again, what is the basis for such a statement? Finally, the authors state “The comparison between the 90°/0° and 0°/90° cases showed that the inner layer plays the major role in ventricle stiffness.” Once again, the basis for such a conclusion is a mystery.

Thank you for your comment. We changed the text as follows: “As shown in Figure 8 (dashed lines), the early development of the endocardial layer resulted in a pressure volume plot similar to the adult-type array, while the inverse plot model (epicardial layer develops first) showed a lower compliance. (line 260-263).

The results found in figure 8 (former figure 9) were based on in-silico model. These data were generated by the computer, based on the variables used to build the model, as stated in the methods. Of course, it is impossible to manipulate myocardial fibers orientation in humans or in animal models to verify different myocardial fiber arrangements.

In summary, unless the authors are willing to present their data in a careful, coherent and thoughtful manner, and unless the authors are better able to justify their insilico model with additional details in describing the result, I cannot be supportive of publication. I note that this paper has been reviewed twice previously and my comments align well with previous critiques.

We hope that we have answered the reviewer's questions in a comprehensive manner. In the last revision, many comments were addressed to histology section and how histological data were used in in-silico model. We hope that, following the reviewer’s suggestion, the paper is more clear, now.

---

## [Decision Letter · Decision Letter 3]

13 Nov 2023

PONE-D-22-00611R3Development of the Fetal Myocardium: The Impact of Fiber Orientation on Left Ventricular MechanicsPLOS ONE

Dear Dr. Castaldi,

Thank you for submitting your manuscript to PLOS ONE. After careful consideration, we feel that it has merit but does not fully meet PLOS ONE’s publication criteria as it currently stands. Therefore, we invite you to submit a revised version of the manuscript that addresses the points raised during the review process.

The paper has again been reviewed by 2 Reviewers.  Reviewer #2 reviewed the last version of the paper and Reviewer #1 is a new Reviewer giving a fresh look at the paper. 

Both Reviewers are critical of the in silico data.  If you are unable to better justify the model calculation, I strongly suggest yu remove this data.  If you choose to continue presenting these findings, you need to thoroughly address the Reviewers comments.

Reviewer #1 also had requests for changes to the sections related to the histology and echo.  Please address those completely. 

It is also noted that the last revised manuscript did not identify the changes made to the manuscript.  Please do this in the following submssion.   

We look forward to receiving your revised manuscript.

Kind regards,

Peter H. Backx

Academic Editor

PLOS ONE

Reviewers' comments:

Reviewer's Responses to Questions

**Comments to the Author**

1. If the authors have adequately addressed your comments raised in a previous round of review and you feel that this manuscript is now acceptable for publication, you may indicate that here to bypass the “Comments to the Author” section, enter your conflict of interest statement in the “Confidential to Editor” section, and submit your "Accept" recommendation.

Reviewer #3: (No Response)

Reviewer #4: (No Response)

2. Is the manuscript technically sound, and do the data support the conclusions?

Reviewer #3: Partly

Reviewer #4: Partly

3. Has the statistical analysis been performed appropriately and rigorously? 

Reviewer #3: I Don't Know

Reviewer #4: Yes

4. Have the authors made all data underlying the findings in their manuscript fully available?

Reviewer #3: No

Reviewer #4: Yes

5. Is the manuscript presented in an intelligible fashion and written in standard English?

Reviewer #3: Yes

Reviewer #4: Yes

6. Review Comments to the Author

Reviewer #3: In my previous review I suggested a number of changes. A number of the suggested changes are inadequately addressed.

Strengths

The Abstract is much improved.

Remaining issues

1. The authors should better (more fully explain) how they aligned the histology measurements with the echo measurements.

2. I still see little point in illustrating the data in Figure 8. Without a better description of what goes into the model and how you incorporated the histological data into the model, along with a careful discussion/explanation of the assumptions, I still think this data/analysis adds little of value to the paper. As a results and as I stated in my last review the basis for conclusions derived from these simulations remains a mystery.

Reviewer #4: The paper studied echo and histology of fetal hearts, and peformed diastolic pressurization on finite element model. There seems to several rounds of review by now. I try here to suggest mostly writing amendments.

1. There is some value in the data in terms of histology measurements for later stages of fetuses, which is lacking in the literature.

- However, the authors should detail the sample sizes of that investigation. How many samples were investigated for each age for both the thickness measurements and fiber orientation measurements?

- The authors provided r and p values for trend over age for thicknesses, but we typically expect a plot of meaurements over time and a best fit line across it. This is important to gain credibility and confidence in the conclusions.

- authors should also state clearly what was their crieteria for differentiating different layers from histology, it doesnt seem quite obvious from the histology images in supplementary figure. How did they choose which tissue is in which layer?

- There is no statistics of trend over age for fiber orientation. This should be attempted if possible. If samples are too low for this, authors should state this. even low sample size data is valuable when there is no other data around.

- The fiber orientation description in lines 231-232 is not turue. the fibers dont go to + 90 or -90 according to figure 6. For a paper that focussed on fiber orientation, not being able to characterize fiber orientation properly will be very negative to a reader/reviewer.

2. The in silico work

- The model is not explained well. what software was used? FeBio? FEniCS? Abaqus? Ideally the simulation settings files should be shared for others to verify. The geometry assumed for the heart in the simulation is very consequential. it could complete change outcomes one way or another. How thick were the walls assumed in the simulations? are they realistic? from figure 7 they look very thin, might no be realistic. Otherwise, from experience, it should be very difficult to get those results in figure 7d, if we have a normal stiffness model and physiologic geometry.

- there should be limitations on how adult canine stiffness model is used here for fetal hearts. Should the fetal heart be less stiff?

- there should be limitation text on how active tension and systole was not simulated. For a proper comparison with echo, you need this, as including systolic behaviour can change comparisons with echo.

- I think there are several sweeping claims that should be removed:

a. line 273, claim that there is correlation between fiber orientation, histology and in vivo functional changes. The simulations did not account for active contraction, and contractility can change things. No formal correlation analysis is done.

b. line 277, histology shows improvement on LV function. how is this demonstrated? strains and most strain rates seems to decrease. How is improved function shown? histology cannot show function, the simualtions lacked active tension, and cannot prove that the histology should lead to increased function.

c. lines 350 - 355, claims that pressure volume plots with different fibers can explain histological changes in heart failure. Authors have not shown fiber orientation change in heart failure, and there can be a confusion between fetal heart investigation here, and heart failure in adult settings as it is. There is no investigation of stiffness, and so authors should not use stiffness to support claims here.

3. the echo measurements

- the measurements seem large from past reviews. but the variability is large (van Oostrum NH, European Journal of Obstetrics & Gynecology and Reproductive Biology. 2020 Jul 1;250:178-87.). The current measurements at the endo layer are beyond the higher end of that review. Given the high variability of measurements out there, this might be ok, but should be explicitly addressed in the text.

- discussion section lines 303-310 seems overly optimistic on speckel tracking for echo strain meaurements. As the cited paper above showed, variability in hands of many different experts lead to highly varying strain results. I think the authors should acknowledge that there is a lot of variability out there, and use that to justify that their strains is higher than others.

Finally: The title of the paper is about myofiber and LV mechanics, but the value of the paper seems to be more on the histology and echo measurements, less on the biomechanics and simulations, given that there's no active tension or systolic simulations. authors could conder address this.

7. PLOS authors have the option to publish the peer review history of their article (what does this mean? ). If published, this will include your full peer review and any attached files.

**Do you want your identity to be public for this peer review?** For information about this choice, including consent withdrawal, please see our Privacy Policy .

Reviewer #3: No

Reviewer #4: No

---

## [Author Response · Author response to Decision Letter 4]

9 Apr 2024

Review Comments to the Author

Reviewer #3: In my previous review I suggested a number of changes. A number of the suggested changes are inadequately addressed.

Strengths

The Abstract is much improved.

Remaining issues

1. The authors should better (more fully explain) how they aligned the histology measurements with the echo measurements.

Thank you for this comment. We added Figure 7 to show the increase of epi/endo thickness over gestational age, and we added a statistical test to compare echo by histology.

Statistical analysis: “The correlation between epicardial to endocardial ratio assessed by speckle tracking and histology was performed by using Spearman’s Rank Test. Data were clusterized for gestational week. Thus, we calculated the mean echocardiographic epi/endo ratio and compared echo data with histological data at any gestational week.”

Results: “We found an excellent correlation between the epicardial to endocardial strain ratio and epicardial to endocardial wall thickness (r=0.950, p<0.001).”

2. I still see little point in illustrating the data in Figure 8. Without a better description of what goes into the model and how you incorporated the histological data into the model, along with a careful discussion/explanation of the assumptions, I still think this data/analysis adds little of value to the paper. As a results and as I stated in my last review the basis for conclusions derived from these simulations remains a mystery.

According with the Editor, we removed in-silico model section from our paper.

Reviewer #4: The paper studied echo and histology of fetal hearts, and peformed diastolic pressurization on finite element model. There seems to several rounds of review by now. I try here to suggest mostly writing amendments.

1. There is some value in the data in terms of histology measurements for later stages of fetuses, which is lacking in the literature.

- However, the authors should detail the sample sizes of that investigation. How many samples were investigated for each age for both the thickness measurements and fiber orientation measurements?

- The authors provided r and p values for trend over age for thicknesses, but we typically expect a plot of meaurements over time and a best fit line across it. This is important to gain credibility and confidence in the conclusions.

Thank you for this comment. We were able to collect about one patient for gestational week. Unfortunately, is difficult to have normal specimens eligible for this kind of study. This is the reason why figure 6 (now Figure 8) doesn’t show SD lines plot.

- authors should also state clearly what was their crieteria for differentiating different layers from histology, it doesnt seem quite obvious from the histology images in supplementary figure. How did they choose which tissue is in which layer?

Thank you for this comment. We further specified in the methods how we measured relative wall thickness. Of course, the contour might be irregular. Thus, multiple measurement were performed.

“The layers were identified according to different orientation of the fibers: vertically cut fibers of middle myocardium appeared lighter in color, because of the major amount of Orange G staining cytoplasm, making it possible to recognize them from the other layers. In the section it was therefore possible to clearly distinguish the longitudinal fibers of the endocardium and epicardium, colored purple red, and the fibers of the mesocardium, which presented a lighter shade as well as a different pattern. Using the same software, we were able to trace line segments for which the program gave us numerical values in micrometer (μm). We measured total wall thickness, non-compacted thickness, endocardial thickness, middle myocardium thickness and epicardial thickness. We carefully avoided artifacts. The measurements were repeated 5 times for each level and in different parts of the specimen. We recorded the mean value for each layer. In addition, the relative layer thickness was calculated.”

- There is no statistics of trend over age for fiber orientation. This should be attempted if possible. If samples are too low for this, authors should state this. even low sample size data is valuable when there is no other data around.

Thank you for this comment. We were able to build the plot available in figure 8, where the reader can see how the angle in the seven layers changes over the gestation. In particular, the epicardial layer is 90° oriented at 13 GW, the second layer (over 7) starts to be longitudinal at the 20th GW, the third layer begins at 30 GW. Thus, we added this point in the limitations.

“Furthermore, the sample size was too small to develop myocardial fiber angulation plots for gestational age. Figure 7 simply represents the value obtained from the single sample at a given gestational age.”

- The fiber orientation description in lines 231-232 is not true. the fibers dont go to + 90 or -90 according to figure 6. For a paper that focused on fiber orientation, not being able to characterize fiber orientation properly will be very negative to a reader/reviewer.

Thank you for this comment. Figure 6 (now figure 8) represents a mean value. If we analyze the most endocardial myocytes, they are 90° oriented. This can be seen also in the ref (Streeter DD, Spotnitz HM, Patel DP, Ross J, Sonnenblick EH. Fiber orientation in the canine left ventricle during diastole and systole. Circ Res 1969;24(3):339–47.)

We better specified this point in the text.

“The angle of the fibers progressively decreased from 90 to 0 from the endocardium to the mesocardium and from 0 to -90° from the mesocardium to the epicardium. We divided myocardium in seven layers, in Figure 7 was shown the mean angle of the single layer. The angle gradient was stable in the endocardial layer, while was steeper in the epicardium at 13 weeks of gestation, reducing the angle gradient proportionally with the thickening of the epicardial layer.”

2. The in silico work

- The model is not explained well. what software was used? FeBio? FEniCS? Abaqus? Ideally the simulation settings files should be shared for others to verify. The geometry assumed for the heart in the simulation is very consequential. it could complete change outcomes one way or another. How thick were the walls assumed in the simulations? are they realistic? from figure 7 they look very thin, might no be realistic. Otherwise, from experience, it should be very difficult to get those results in figure 7d, if we have a normal stiffness model and physiologic geometry.

- there should be limitations on how adult canine stiffness model is used here for fetal hearts. Should the fetal heart be less stiff?

- there should be limitation text on how active tension and systole was not simulated. For a proper comparison with echo, you need this, as including systolic behaviour can change comparisons with echo.

- I think there are several sweeping claims that should be removed:

a. line 273, claim that there is correlation between fiber orientation, histology and in vivo functional changes. The simulations did not account for active contraction, and contractility can change things. No formal correlation analysis is done.

b. line 277, histology shows improvement on LV function. how is this demonstrated? strains and most strain rates seems to decrease. How is improved function shown? histology cannot show function, the simualtions lacked active tension, and cannot prove that the histology should lead to increased function.

c. lines 350 - 355, claims that pressure volume plots with different fibers can explain histological changes in heart failure. Authors have not shown fiber orientation change in heart failure, and there can be a confusion between fetal heart investigation here, and heart failure in adult settings as it is. There is no investigation of stiffness, and so authors should not use stiffness to support claims here.

According with the Editor, we removed this part of the paper.

3. the echo measurements

- the measurements seem large from past reviews. but the variability is large (van Oostrum NH, European Journal of Obstetrics & Gynecology and Reproductive Biology. 2020 Jul 1;250:178-87.). The current measurements at the endo layer are beyond the higher end of that review. Given the high variability of measurements out there, this might be ok, but should be explicitly addressed in the text.

We used a layer specific software; thus, the region of interest was smaller, this is the reason why the endocardial layer showed a slightly higher SD. We already stated this in limitations, however, we better specified this point.

“Thus, the standard deviation was higher than values calculated in full-thickness strain values.”

- discussion section lines 303-310 seems overly optimistic on speckel tracking for echo strain meaurements. As the cited paper above showed, variability in hands of many different experts lead to highly varying strain results. I think the authors should acknowledge that there is a lot of variability out there, and use that to justify that their strains is higher than others.

We partially agree on this. It’s true that layer specific strain is more complex. However, “global wall” strain is more robust, when acoustic window is adequate. As mentioned above, we added in the limitations the higher variability of layer specific strain.

Finally: The title of the paper is about myofiber and LV mechanics, but the value of the paper seems to be more on the histology and echo measurements, less on the biomechanics and simulations, given that there's no active tension or systolic simulations. authors could conder address this.

We agree that the title should change now: “Development of the Fetal Myocardium and Changes in Myocardial Fibers Orientation”

---

## [Decision Letter · Decision Letter 4]

6 Aug 2024

PONE-D-22-00611R4Development of the Fetal Myocardium and Changes in Myocardial Fibers OrientationPLOS ONE

Dear Dr. Castaldi,

Thank you for submitting your manuscript to PLOS ONE. After careful consideration, we feel that it has merit but does not fully meet PLOS ONE’s publication criteria as it currently stands. Therefore, we invite you to submit a revised version of the manuscript that addresses the points raised during the review process.

**Here are the Comments from the reviewer:** :

**Reviewer 3** has asked for minor changes in the manuscript.

The authors examine the developmental changes in LV myocardial structure (histology) and function (speckle tracking) during fetal development. The expected progressive changes in both structure and function were observed. The authors conclude that “Left ventricular myocardium maturation begins early during fetal life and starts from the differentiation of a subendocardial layer. The development of the epicardial layer is slower and it is completed late in fetal life.”

The paper has been improved and simplified by removal of the simulation data. I support publication provided the final adjustment are made.

1. I believe that the term “differentiation” that is used in the conclusion is not very informative. The conclusion would better read (and by more accurate) by stating: “As the fetus develops, both the relative tissue volume and peak systolic strain rates shift together from the endocardium towards the epicardium”.

2. To measure strain, the myocardium was automatically divided into endocardial and epicardial layers.

What does the term “automatically” mean? Please provide details.

3. The authors state “All samples were included in paraffin, cut into 5 µm slices, and then stained with Azan-Mallory for measurement of layer thickness and myofibers orientation.” At this thickness, the term “all” implies thousands of slices. Do the authors mean that several transverse slices were examined from “all” the fetal hearts examined?

4. The sentence lacks a verb. “As expected, wall thickness and diameters while the left ventricular volume increased based on a cubic regression model (Figure 5). “

5. The word “determinate” is an adjective not a verb. I think the word “suggests” might be considered.

6. The statement “These findings may be at least in part responsible of the lower tolerance to pre-load and after-load variation in preterm newborns.” I do not understand this conclusion.

**Reviewer 4** has accepted the manuscript for publication but has dropped few suggestions that needs to be addressed.

I thank the authors for responding well. Removing the simulations was a very good move. The focus on the echo and histology measurements is better now, and it covers a gap in the literature. I recommend publication of the article.

I just want to point out that the last paragraph of introduction stated the objective differently from the title of the paper. it stated quantifying thicknesses only, and did not mention fiber orientations.

May I further suggest to the authors that one reason for the differential strains between epi and endo is likely due to the the curved surface of the heart and the radial expansion during systole. here's a reference to check out:

Ren et al., Journal of the American Society of Echocardiography. 2023 May 1;36(5):543-52.

We look forward to receiving your revised manuscript.

Kind regards,

Dhruba Shrestha, MD

Academic Editor

PLOS ONE

Journal Requirements:

Additional Editor Comments:

There are minor changes to be made before deciding for publication. After going through the article, I have one query about the speckle tracking of fetal myocardium. Since this is a very new concept, is there any standard values for the fetus? How is it possible to do a statistical analysis if there are no standard values.

Reviewers' comments:

Reviewer's Responses to Questions

**Comments to the Author**

1. If the authors have adequately addressed your comments raised in a previous round of review and you feel that this manuscript is now acceptable for publication, you may indicate that here to bypass the “Comments to the Author” section, enter your conflict of interest statement in the “Confidential to Editor” section, and submit your "Accept" recommendation.

Reviewer #3: (No Response)

Reviewer #4: (No Response)

2. Is the manuscript technically sound, and do the data support the conclusions?

Reviewer #3: Yes

Reviewer #4: Yes

3. Has the statistical analysis been performed appropriately and rigorously? 

Reviewer #3: I Don't Know

Reviewer #4: Yes

4. Have the authors made all data underlying the findings in their manuscript fully available?

Reviewer #3: Yes

Reviewer #4: Yes

5. Is the manuscript presented in an intelligible fashion and written in standard English?

Reviewer #3: No

Reviewer #4: Yes

6. Review Comments to the Author

Reviewer #3: The authors examine the developmental changes in LV myocardial structure (histology) and function (speckle tracking) during fetal development. The expected progressive changes in both structure and function were observed. The authors conclude that “Left ventricular myocardium maturation begins early during fetal life and starts from the differentiation of a subendocardial layer. The development of the epicardial layer is slower and it is completed late in fetal life.”

The paper has been improved and simplified by removal of the simulation data. I support publication provided the final adjustment are made.

1. I believe that the term “differentiation” that is used in the conclusion is not very informative. The conclusion would better read (and by more accurate) by stating: “As the fetus develops, both the relative tissue volume and peak systolic strain rates shift together from the endocardium towards the epicardium”.

2. To measure strain, the myocardium was automatically divided into endocardial and epicardial layers.

What does the term “automatically” mean? Please provide details.

3. The authors state “All samples were included in paraffin, cut into 5 µm slices, and then stained with Azan-Mallory for measurement of layer thickness and myofibers orientation.” At this thickness, the term “all” implies thousands of slices. Do the authors mean that several transverse slices were examined from “all” the fetal hearts examined?

4. The sentence lacks a verb. “As expected, wall thickness and diameters while the left ventricular volume increased based on a cubic regression model (Figure 5). “

5. The word “determinate” is an adjective not a verb. I think the word “suggests” might be considered.

6. The statement “These findings may be at least in part responsible of the lower tolerance to pre-load and after-load variation in preterm newborns.” I do not understand this conclusion.

Reviewer #4: I thank the authors for responding well. Removing the simulations was a very good move. The focus on the echo and histology measurements is better now, and it covers a gap in the literature. I recommend publication of the article.

I just want to point out that the last paragraph of introduction stated the objective differently from the title of the paper. it stated quantifying thicknesses only, and did not mention fiber orientations.

May I further suggest to the authors that one reason for the differential strains between epi and endo is likely due to the the curved surface of the heart and the radial expansion during systole. here's a reference to check out:

Ren et al., Journal of the American Society of Echocardiography. 2023 May 1;36(5):543-52.

7. PLOS authors have the option to publish the peer review history of their article (what does this mean? ). If published, this will include your full peer review and any attached files.

**Do you want your identity to be public for this peer review?** For information about this choice, including consent withdrawal, please see our Privacy Policy .

Reviewer #3: No

Reviewer #4: No

---

## [Author Response · Author response to Decision Letter 5]

18 Sep 2024

Reviewer 3 has asked for minor changes in the manuscript.

The authors examine the developmental changes in LV myocardial structure (histology) and function (speckle tracking) during fetal development. The expected progressive changes in both structure and function were observed. The authors conclude that “Left ventricular myocardium maturation begins early during fetal life and starts from the differentiation of a subendocardial layer. The development of the epicardial layer is slower and it is completed late in fetal life.”

The paper has been improved and simplified by removal of the simulation data. I support publication provided the final adjustment are made.

1. I believe that the term “differentiation” that is used in the conclusion is not very informative. The conclusion would better read (and by more accurate) by stating: “As the fetus develops, both the relative tissue volume and peak systolic strain rates shift together from the endocardium towards the epicardium”.

Thank you for the suggestion, we modified either in the abstract and in the conclusion.

2. To measure strain, the myocardium was automatically divided into endocardial and epicardial layers.

What does the term “automatically” mean? Please provide details.

Automatically means that it was not manually adjustable/modifiable. Echopac does not allow to modify the relative thickness of the layers, it just divides in inner and outer. Other software (i.e. Tomtec) permit to modify manually this parameter. Anyway, we specified what is manually modifiable and what not in methods.

3. The authors state “All samples were included in paraffin, cut into 5 µm slices, and then stained with Azan-Mallory for measurement of layer thickness and myofibers orientation.” At this thickness, the term “all” implies thousands of slices. Do the authors mean that several transverse slices were examined from “all” the fetal hearts examined?

Thank you for the comment. All might be misleading, we modified “The samples were included in paraffin…”. All means that all the samples were managed in the same way.

4. The sentence lacks a verb. “As expected, wall thickness and diameters while the left ventricular volume increased based on a cubic regression model (Figure 5). “

Sorry, probably, the sentence was accidentally truncated. The complete sentence was:

“As expected, wall thickness and diameters grew following a linear regression model, while the left ventricular volume increased based on a cubic regression model”

5. The word “determinate” is an adjective not a verb. I think the word “suggests” might be considered.

Thank you, we changed accordingly.

6. The statement “These findings may be at least in part responsible of the lower tolerance to pre-load and after-load variation in preterm newborns.” I do not understand this conclusion.

Several studies demonstrated that LV deformation properties are impaired in premature newborns compared to normal (at term) neonates. On the other hand, extreme premature newborns show a reduced tolerance to pre-load and after-load variations (most commonly seen in PDA – low systemic output for steal before, post ligation syndrome after closure, evidenced only in prematures and not in older infants), as well as a reduced response to inotropes. Based on our study, and on similar studies in premature newborns, these clinical evidence might be, at least in part, justified by an incomplete myocardial development in these subjects.

Reviewer 4 has accepted the manuscript for publication but has dropped few suggestions that needs to be addressed. I thank the authors for responding well. Removing the simulations was a very good move. The focus on the echo and histology measurements is better now, and it covers a gap in the literature. I recommend publication of the article.

I just want to point out that the last paragraph of introduction stated the objective differently from the title of the paper. it stated quantifying thicknesses only, and did not mention fiber orientations.

Thank you for the suggestion, we better specified in the aim this point.

May I further suggest to the authors that one reason for the differential strains between epi and endo is likely due to the the curved surface of the heart and the radial expansion during systole. here's a reference to check out: Ren et al., Journal of the American Society of Echocardiography. 2023 May 1;36(5):543-52.

We completely agree. Based on previous reviewers suggestions, we removed the in-silico model analysis, where, at least in part, we described this point. It is well known that strain depends not just on the relative angulations but also on the shape of the ventricle and on the curvature of the surface. Thus, we did not specified further in this paper. This is the reason why endocardial strain paradoxically decreases over the time (L/T ratio decreases over the time, in particular in the second trimester of gestation) and epi/endo strain ratio increases, accordingly with the epi/endo histologic relative wall thickness.

---

## [Editor Report · Decision Letter 5]

29 Nov 2024

Development of the Fetal Myocardium and Changes in Myocardial Fibers Orientation

PONE-D-22-00611R5

Dear Dr. Castaldi,

We’re pleased to inform you that your manuscript has been judged scientifically suitable for publication and will be formally accepted for publication once it meets all outstanding technical requirements.

Kind regards,

Dhruba Shrestha, MD

Academic Editor

PLOS ONE
---

## [Editor Report · Acceptance letter]

PONE-D-22-00611R5

PLOS ONE

Dear Dr. Castaldi,

I'm pleased to inform you that your manuscript has been deemed suitable for publication in PLOS ONE. Congratulations! Your manuscript is now being handed over to our production team.

Kind regards,

on behalf of

Dr. Dhruba Shrestha

Academic Editor

PLOS ONE